# Groundwater Seepage Modeling in a River-Canal System based on Physics-Informed Neural Networks

## Abstract

Neural networks, especially deep learning, have achieved revolutionary advances in several domains, including image and speech recognition, with excellent results. However, their reliance on labeled data, lack of interpretability, and inconsistency with physical principles limit their applicability in groundwater seepage prediction and other scientific disciplines. Physics-Informed Neural Networks (PINNs) significantly improve these issues by integrating physical knowledge with neural networks. This study focuses on modeling the groundwater flow field and proposes a physics-informed river-canal groundwater seepage model (PI-RGSM). This model enables self-supervised learning by incorporating hard constraints of boundary and initial conditions, utilizing hydrogeological parameters and boundary conditions as direct inputs, thus diminishing dependence on observable data. Compared to the baseline PINNs, the PI-RGSM adapts to and accurately predicts diverse seepage situations with just one training session, achieving a mean coefficient of determination of 0.978. To further enhance applicability in complex dynamic groundwater seepage situations, we propose PI-RGSM-K, which builds upon PI-RGSM. This model simulates heterogeneous groundwater seepage fields and improves performance in complex seepage environments through parameterized hydraulic conductivity field $K(x, y)$ and fine-adjusted model architecture, attaining a mean coefficient of determination of 0.982. The physics-informed neural network models proposed in this study demonstrate exceptional efficacy in precisely forecasting groundwater seepage behavior.

## 1 Introduction

Groundwater is a crucial source of fresh water for the global population, essential for residential life, agricultural irrigation, industrial production, and ecosystem protection (Massuel et al., 2018; Xie et al., 2023; Mishra, 2023). It is imperative to manage this rich resource sustainably to ensure its long-term availability. Therefore, it is very important to understand groundwater dynamics. Field-based observations, theoretical analysis, hydrogeological modeling are essential ways to study groundwater dynamics (Li et al., 2021; Su et al., 2023). However, although some field-based observations (e.g., pump tests, tracer Tests, monitoring wells, etc.) are inevitable, large amount of field-based observations for analyzing groundwater dynamics are time-consuming and labor-intensive; although theoretical analysis is effective in providing deep understanding of the basic principles and flexibility in exploring simplified complex systems, the oversimplification, heavy dependence on assumptions and lack of specificity limit the further application. Hydrogeological modeling is widely used to simulate groundwater dynamics for its relatively easy and cheap of implementing, ability of complex system representation, scenario analyzing, results visualization (Chen et al., 2020; Jing et al., 2023).

With the evolution of scientific paradigms (Caíno-Lores et al., 2020), researches on natural science has mitigated from empirical and theoretical approaches to computational and data-intensive scientific paradigms driven by computer simulations and big data (Babovi & Bajat, 2023; Li et al., 2023). In the field of computational fluid dynamics, numerical models have achieved accurate simulations of physical processes (refer to Appendix A). Important issues in those spatial discretization methods for solving the partial differential equations (PDEs) of groundwater are the computationally intensive. Continuous advancements in data-intensive scientific paradigms have empowered deep

neural networks (DNN) to achieve significant breakthroughs in many fields (refer to Appendix A). However, the "black box" nature of DNN exhibiting a lack of transparency in their decision-making processes and the significant dependence on extensive training data, limit their use in groundwater research. The physics-informed neural networks (PINNs) which combine the physical laws and neural networks have improved the interpretability of models and diminished reliance on extensive datasets. Wang et al. (2020) proposed a theoretically guided neural network model (TgNN) based on the loss function of groundwater seepage differential equations, which improved the generalization performance of the groundwater seepage model with limited observational data. Additionally, the team demonstrated that the physics-informed model could effectively use physical information to predict groundwater seepage responses beyond the training set through groundwater seepage transfer learning tasks. Wang et al. (2021a) proposed a neural network constrained by geostatistical information and a physics-guided autoencoder based on convolutional neural networks (CNNs) (Wang et al., 2021b), applying the physical guidance method to the parameter inversion of groundwater hydraulic conductivity fields. The model's accuracy and practicality were further enhanced. The study by (Pashaei Kalajahi et al., 2022) demonstrated that, even with limited data, physics-guided neural networks could accurately estimate key parameters such as hydraulic conductivity and porosity of the medium, showcasing their strong ability to combine data-driven approaches with physical guidance. Daolun et al. (2021) established a new specialized neuron model incorporating pressure gradient information and proposed an algorithm called signpost neural network (SNN), which significantly improves the accuracy of solving unsteady seepage partial differential equations. Although significant progress has been made in improving groundwater seepage models using PINNs, current models still heavily depend on the adequacy and quality of observed data. Moreover, after the model has been trained, its applicability is generally limited to specific hydrogeological parameter settings, making it difficult to generalize to broader or unforeseen hydrogeological scenarios, thus restricting the model's versatility and practicality.

**Contributions:** In this paper, a self-supervised deep learning method for groundwater seepage prediction, PI-RGSM, is proposed to address the limitations of baseline PINNs, which depends on observed data and have limited flexibility. Strategies of incorporating hard constraints combined with boundary condition inputs into the baseline PINNs and utilizing hydrogeological parameters and boundary conditions as direct inputs are adopted. These allow the model to adapt new hydrogeological environments without retraining. Additionally, an extended version of the model, PI-RGSM-K, is developed to simulate more complex heterogeneous seepage scenarios by parameterizing hydraulic conductivity field $K(x, y)$. This enhances the precision and adaptability of groundwater seepage prediction, effectively overcoming the limitations of baseline models in heterogeneous seepage fields. Several representative river-canal groundwater seepage cases are designed to evaluate the accuracy and efficiency of the proposed method.

**Paper Organization:** The theoretical background is summarized in Section 2, along with a discussion of the improvement method for baseline PINNs. Section 3 introduces the construction and experimental setup of the PI-RGSM and PI-RGSM-K models. A series of experiments verifying the predictive capability of PI-RGSM and its extended model, PI-RGSM-K, under various hydrogeological conditions (including variations in precipitation, hydraulic conductivity, boundary conditions, and heterogeneous hydraulic conductivity field) are presented in Section 4. Finally, the summary and conclusions are provided in Section 5.

## 2 METHOD

### 2.1 GROUNDWATER SEEPAGE IN A RIVER-CANAL SYSTEM

As a key hydraulic engineering structure, river-canal system is a typical system which combines river, canal and groundwater. The river-canal system significantly influences the surrounding groundwater flow, making its study crucial for groundwater resource assessment, artificial drainage system design, and agricultural irrigation strategy formulation. Its uniqueness lies in considering scenarios where two parallel river-canals cross the aquifers to reach impermeable bottom layers. As illustrated in Figure 1, the groundwater between the two river-canals forms a specific flow pattern, with the left and right canals serving as Dirichlet boundaries to create a convex hydraulic head surface, known as the phreatic surface. The morphology of the phreatic line, represented by the cross-sectional curve of the phreatic hydraulic head $H$, is affected by the water levels $H_a$ and $H_b$ on both sides, as well as rainfall infiltration $W$. In a phreatic aquifer, assuming negligible variation in hydrogeological conditions in the vertical direction, groundwater flow is governed by the two-dimensional (in the

horizontal plane $XY$) seepage PDE (Equation 1). Where $t$ is time ($T$); $K$ is the hydraulic conductivitie ($LT^{-1}$) along the $x$, $y$ axes, respectively. $\mu_s$ is the specific storage ($L^{-1}$) of the aquifer below the free surface. Full details in Appendix B.

$$\mu\frac{\partial H}{\partial t} = \frac{\partial}{\partial x}\left(KH\frac{\partial H}{\partial x}\right) + \frac{\partial}{\partial y}\left(KH\frac{\partial H}{\partial y}\right) + W \tag{1}$$

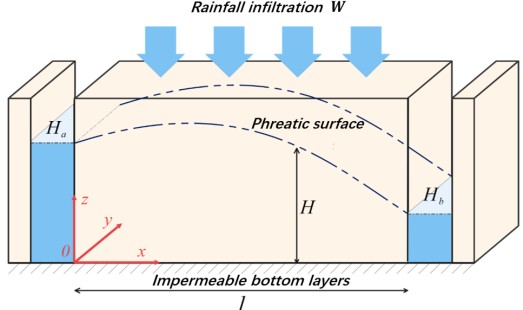

Figure 1: The schematic diagram of groundwater seepage in a river-canal system.

## 2.2 Loss function of baseline PINNs

In the equations of physics-informed groundwater seepage, the loss function is integrated with the basic PDE of groundwater seepage, initial and boundary conditions, observational data, and engineering control to guide the optimization direction of the neural network and ensure its physical consistency.

(1) **Loss function of PDE:** Considering the specific case of a two-dimensional heterogeneous seepage field with unit formation thickness, the seepage equation is:

$$\frac{\partial}{\partial x}\left(K(x,y)H\frac{\partial H(x,y,t)}{\partial x}\right) + \frac{\partial}{\partial y}\left(K(x,y)H\frac{\partial H(x,y,t)}{\partial y}\right) + W = \mu_s\frac{\partial H(x,y,t)}{\partial t} \tag{2}$$

Where $K(x,y)$ is the hydraulic conductivity. Using this equation, we define the PDE residuals $RES_{PDE}$:

$$RES_{PDE} = \frac{\partial}{\partial x}\left(K(x,y)\hat{H}\frac{\partial \hat{H}}{\partial x}\right) + \frac{\partial}{\partial y}\left(K(x,y)\hat{H}\frac{\partial \hat{H}}{\partial y}\right) + W - \mu_s\frac{\partial \hat{H}}{\partial t} \tag{3}$$

When the physical consistency between the input and output variables of the model is higher, $RES_{PDE}$ approaches 0. During the calculation process, $RES_{PDE}$ requires the model output (hydraulic head $\hat{H}$) to perform higher-order differentiation with respect to the input spatial-temporal coordinate points ($x$, $y$, $t$). This task can be performed directly and efficiently using the automatic differentiation provided by modern machine learning frameworks such as TensorFlow or PyTorch. The hyperbolic tangent function (Tanh) is chosen as the activation function in this study because its favorable second-order differentiability meets the model's requirements.

(2) **Constraint from definite solution condition:** Consider the first type boundary condition (Dirichlet condition), where the water level at any given time at every point in a specific boundary region is known, which can be expressed as:

$$H(x,y,t)\big|_{\Gamma_1} = \varphi(x,y,t), \quad (x,y,t) \in \Gamma_1 \tag{4}$$

Where $H(x,y,t)$ represents the hydraulic head at boundary $\Gamma_1$, and $\phi(x,y,t)$ represents the hydraulic head function with spatial-temporal coordinates as the independent variable. By moving the function $\phi(x,y,t)$ to the left-hand side of the equation and substitute the hydraulic head on the boundary by the outputs of neural network ($f_{net}$), the residual of boundary condition $RES_{BC}$ is:

$$RES_{BC} = \hat{H} - \varphi(x,y,t), \quad (x,y,t) \in \Gamma_1 \tag{5}$$

For initial conditions, this can be expressed as $\phi(x,y,0) = H_0(x,y)$, where $H_0$ is the hydraulic head distribution function at the initial time. Therefore, the residual of initial condition $RES_{IC}$ can be expressed as:

$$RES_{IC} = \varphi(x,y,0) - \hat{H}, \quad t = 0 \tag{6}$$

(3) **Constraint from engineering control:** It is often necessary to incorporate specific ecological

and policy-oriented engineering control measures when solving actual groundwater seepage problems. For example, in ecological restoration, it may be essential to limit the decline of the hydraulic head in an area to a predetermined threshold $H_{EC}$. By constructing a residual term ($RES_{EC}$) for engineering control, it is possible to measure the discrepancy between the model prediction and the actual control objective:

$$RES_{EC} = \begin{cases} H_{EC} - \hat{H}, & H_{EC} > \hat{H} \\ 0, & H_{EC} \leq \hat{H} \end{cases} \tag{7}$$

(4) **Constraint from observed data:** The difference between the hydraulic head predicted by the neural network and the actual hydraulic head observed can be quantified by constructing the residual $RES_{OB}$ of the observed data:

$$RES_{OB} = H_{OB} - \hat{H} \tag{8}$$

The combined physics-informed loss function is synthesized from Eq. 3,5,6,7,and 8, to impose multiple constraints on the neural network:

$$\mathcal{L} = \sum_{i=\{PDE,BC,IC,EC,OB\}} \lambda_i \left( \frac{1}{N} \sum_{j=1}^{N} [RES_j]^2 \right)_i \tag{9}$$

Where $j$ represents each input data point. By assigning weights $\lambda_i$ to each residual term, the contribution of different residual constraints to the total loss function can be adjusted to effectively inform the training process of the neural network.

### 2.3 Constraints from boundary and initial condition (hard constraints)

The optimization process of the neural network relies on the physics-informed loss function, which is centered around the groundwater seepage PDE, as discussed in Section 2.2. The physics-informed loss function involves the interaction between multiple loss function terms, and if multiple losses are solved iteratively without additional processing, the solution mode is referred to as a "soft constraint". In soft constraint, the direction of the neural network solution may deviate due to different solution gradients of tasks, particularly when the boundaries of the objective function or initial conditions are unconventional. Thus, assigning weights to the loss function for each task is a relative compromise; however, these loss weights are usually manually adjusted in a tedious trial-and-error process which also increases the computational cost. To address this issue, we introduce the "hard constraints" by constructing the neural network output in the form of an elementary function that satisfies Dirichlet boundary conditions, along with the boundary condition equation. Thus, avoiding directly constructing the residual terms $RES_{BC}$ and $RES_{IC}$ for the boundary and initial conditions, significantly reducing the number of residuals, alleviating the pressure on the model when faced with multiple optimization problems, and facilitating the manual adjustment of training hyperparameters $\lambda_i$. Simultaneously, since the hydraulic head processed with hard constraints satisfies the constraints of boundary and initial conditions, the model does not need to find a specific solution under uncertain boundary conditions during the optimization stage, thus enhancing the overall convergence effect and eliminating the need for labeled data to provide specific solution information.

$$\hat{H}_C = C(x,y,t)\hat{H} + \varphi(x,y,t), \begin{cases} \varphi(x,y,0) = H_0(x,y) \\ \varphi(x,y,t) = H(x,y,t)\big|_{\Gamma_1}, \quad (x,y,t) \in \Gamma_1 \end{cases} \tag{10}$$

Where $\hat{H}_C$ is the predicted hydraulic head of the model considering hard constraints; $C(x,y,t)$ is a boundary constraint function characterized by its output gradually tending toward 0 as the input spatial-temporal coordinates $(x,y,t)$ approach the model boundary or initial time. For groundwater dynamics in the river-canal system addressed in this study, since Dirichlet boundary conditions exist only on the left-hand and right-hand sides, the following constraint function can be constructed:

$$C(x,y,t) = \frac{(x - x_{\min})(x_{\max} - x)t}{N_s} \tag{11}$$

Where $x_{min}$ and $x_{max}$ are the minimum and maximum values of the spatial coordinates in the $x$-axis direction of the canal (the $x$-coordinate values of the left and right boundary conditions); $N_s$ is a zoom factor used to balance the data magnitude, which is set as the maximum value of $l^2 t$. With hard constraints applied, the model's output at boundary and initial points directly matches $\varphi(X)$, while other points are constrained by the PDE. Since $\varphi(X)$ inherently satisfies boundary and initial conditions, separate loss functions for these conditions are unnecessary, reducing the overall loss functions and improving the model's solution accuracy.

## 2.4 INPUT FEATURE FUSION

In the river-canal groundwater seepage field, the water levels $(H_a, H_b)$ of the left and right canals play a crucial role in the morphology of the phreatic surface, and these water levels are the key boundary conditions for defining specific solutions of the PDE of groundwater seepage. In the baseline PINNs (as discussed in Section 2.2), the convergence of the model depends on fixed boundary conditions to find a specific solution, making it difficult to adjust once the boundary conditions are set. In this study, we proposed to adopt an innovative constraint to the equation which integrates the left and right canal water levels $(H_a, H_b)$ as dynamic input features into the model and utilizes them in the constraint equation, thus limiting the model outputs to a preset boundary condition:

$$\hat{H}_C = C(x,y,t)\hat{H}(x,y,t,H_a,H_b) + \varphi_{BC}(x,y,t,H_a,H_b), \begin{cases} \varphi_{BC}(x=x_{\min},y,t) = H_a \\ \varphi_{BC}(x=x_{\max},y,t) = H_b \end{cases} \quad (12)$$

Where $\varphi_{BC}$ is a condition function of definite solution for input features( discussed in Section 2.3) .

In the groundwater seepage equation of a river-canal system(Eq. 2), in addition to the spatial-temporal coordinate variables $(x,y,t)$, the hydraulic conductivity $(K)$ and the source/sink term $(W)$ are also important parameters affecting groundwater seepage behavior. Incorporating these parameters into the model's input features can provide complete hydrogeological background information for the groundwater seepage equation, thereby improving the model's flexibility and adaptability in simulating different groundwater seepage scenarios. In this study, randomly generated hydraulic conductivities and source/sink terms are added to the training dataset of spatial-temporal coordinate points, creating a dataset with more features to integrate hydraulic conductivity and source/sink term information into the training process. These additional variables are then introduced into the physics-informed residual term of PDE (Eq. 13), allowing the model to receive and process diverse seepage scenarios described by the hydrogeological parameters. By constructing random datasets, the model can be trained on a large number of generated training cases, each representing a specific hydrogeological condition, thus enhancing the model's ability to simulate different groundwater seepage conditions.

$$RES_{PDE}(x,y,t,W,K) = \frac{\partial}{\partial x}\left(K\hat{H}_C\frac{\partial \hat{H}_C}{\partial x}\right) + \frac{\partial}{\partial y}\left(K\hat{H}_C\frac{\partial \hat{H}_C}{\partial y}\right) + W - \mu\frac{\partial \hat{H}_C}{\partial t} \quad (13)$$

This model can simulate more complex and diverse hydrogeological conditions by incorporating the hydraulic conductivity, source/sink term, and boundary conditions (left and right canal water levels) into the model input features, covering a wider range of practical applications, greatly enhancing the usefulness and flexibility of the model.

## 3 EXPERIMENT

The PI-RGSM model proposed in this study consists of fully connected neural networks (FCNN), hard constraints, and loss function design. As shown in Figure 2 (ignoring the red dashed line in the input feature section), the input features of the model include the spatial-temporal coordinates of the river-canals seepage $(x,y,t)$, adjustable water levels of the river-canals $(H_a, H_b)$, and hydrogeological parameters (e.g., the source/sink term $W$ and hydraulic conductivity $K$). These multidimensional input data are processed through the neural network to output the initial predicted hydraulic head $H$. The model introduces boundary condition function $C(X)$ and condition function of definite solution $\varphi(X)$ as hard constraints to constrain the model to predict the hydraulic head $(\hat{H}_C)$. Automatic differentiation (AD) is then employed to compute the partial derivatives of hydraulic head. The source/sink term $(W)$ and hydrogeological parameters (e.g., hydraulic conductivity) are incorporated into the PDE constraint $(LOSS_{PHY})$, along with the engineering control loss $(LOSS_{EC})$, to form a system of PDEs (Ketkar et al., 2021). These two loss terms are balanced to optimize the overall training by adjusting the parameters $\lambda_1$ and $\lambda_2$. The model demonstrates flexibility by outputting the corresponding hydraulic head for different boundary conditions and hydrogeological parameters without retraining.

To verify the performance of PI-RGSM, a case involving groundwater seepage in a river-canal system under a unconfined aquifer was established. Without loss generality, the unit of length is set to meter $(m)$, and the unit of time is set to day $(d)$. The schematic of this case is illustrated in Figure 1, where the porous media region between the left and right canals represents the infiltration area. The infiltration area has a length $(x)$ of $40m$ and a width $(y)$ of $10m$. The impermeable bottom boundary is considered to be flat, with a formation thickness $(M)$ is $5m$. The specific storage $(\mu)$ is 0.1. The

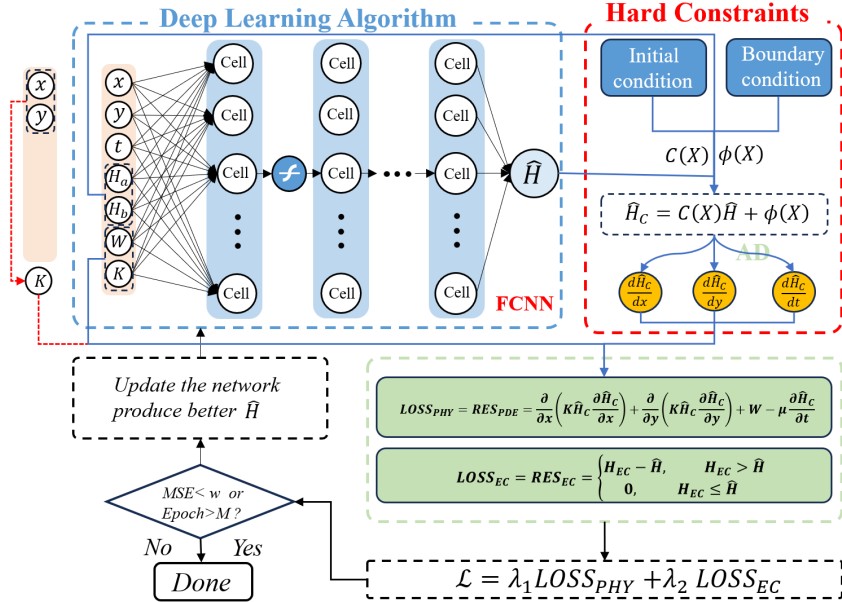

Figure 2: The structure of PI-RGSM and PI-RGSM-K: where the blue line section represents the PI-RGSM, and the red dashed section highlights the extended model PI-RGSM-K, which builds on PI-RGSM by parameterizing $K$ as a spatial function of $x$ and $y$.

hydraulic conductivity ($K$) ranges from $0.5m/d$ to $0.9m/d$. The source/sink term ($W$) is positive (representing recharge) and varies from $0.000m/d$ to $0.007m/d$. The initial water levels ($H_a, H_b$) of the left and right canals vary between $2m$ and $3.5m$.

The core architecture of PI-RGSM is a deep neural network with 7 layers, each containing 50 neurons, using the Adam optimizer. This configuration was chosen to align with the optimal architecture identified during the hyperparameter tuning process for the baseline PINNs (as detailed in the appendix F). According to the experimental settings of hydrogeological parameters, boundary and initial conditions (hard constraints), and engineering control, the physics-informed losses ($LOSS_{PHY}$ and $LOSS_{EC}$) are defined as follows:

$$\begin{cases} LOSS_{PHY} = RES_{PDE} = \dfrac{\partial}{\partial x}\left(K\hat{H}_C \dfrac{\partial \hat{H}_C}{\partial x}\right) + \dfrac{\partial}{\partial y}\left(K\hat{H}_C \dfrac{\partial \hat{H}_C}{\partial y}\right) + W - 0.1\dfrac{\partial \hat{H}_C}{\partial t} \\[4mm] LOSS_{EC} = RES_{EC} = \begin{cases} 2 - \hat{H}_C, & \hat{H}_C < 2 \\ 0, & 2 \le \hat{H}_C \end{cases} \end{cases} \quad (14)$$

Since the PI-RGSM model adopts a self-supervised learning approach which does not rely on labeled training data. It requires the generation of random values within specified ranges for each input feature to simulate diverse scenarios due to the scarcity of real hydrogeological data in groundwater systems. Table 1 lists all input features and their corresponding value ranges. These datas were validated through their application in professional simulation software like MODFLOW, ensuring their relevance to real-world conditions. In this experiment, 10,000 training samples were generated, and all input data were normalized. The model was trained using these data for 1,000 iterations.

Table 1: Input features and value ranges of the PI-RGSM

| Input features | $x(m)$ | $y(m)$ | $t(d)$ | $W(m/d)$ | $K(m/d)$ | $H_a(m)$ | $H_b(m)$ |
|---|---|---|---|---|---|---|---|
| Value ranges | [0,40] | [0,10] | [0,100] | [0,0.007] | [0.5,0.9] | [2,3.5] | [2,3.5] |

Although the PI-RGSM model demonstrates potential in predicting groundwater seepage in river-canal system, its application is limited by relatively simplified geological environment assumptions, such as the homogeneous hydraulic conductivity which ignores the heterogeneity of the hydraulic conductivity in practical application scenarios. To address complex geological conditions, this study proposes an extension of the PI-RGSM framework (hereafter refer to PI-RGSM-K).

To consider the heterogeneity of the hydraulic conductivity field, we replace the homogeneous hydraulic conductivity (a single value for the whole area) as input by regarding the hydraulic conduc-

tivity as a function of the two-dimensional space coordinates $x$ and $y$ (i.e., $K(x, y)$). This approach allows the model to directly map the hydraulic conductivities and their gradients along the coordinate axes into a physics-informed loss function, thus incorporating geological features in training process. Based on this improvement, the core structure of the PI-RGSM-K can be directly extended from the PI-RGSM framework (shown in Figure 2). The key modification involves constructing the hydraulic conductivity $K$ as an independent function $K(x, y)$ dependent on two-dimensional spatial coordinates. Along with the source/sink term $W$ in the input features, it is incorporated into the PDE constraint ($LOSS_{PHY}$) to improve prediction under heterogeneous porous media. Compared to PI-RGSM, PI-RGSM-K uses a function $K$=-0.01$x$+0.8 as input (data details in appendix E) in our study. However, $K$ can also be parameterized using other functions of $x$ and $y$, or even as a randomly generated hydraulic conductivity field. When $K$ is a random permeability field that can better reflect the real geological environment, we can build a separate neural network to take $(x, y)$ as the input of the neural network, and $K$ as the output. Using the powerful function approximation ability and automatic differential characteristics of the neural network, we can obtain the gradient characteristics of $K$ about $x$ and $y$, which is the direction we need to improve in the future.

## 4 RESULTS AND DISCUSSION

### 4.1 PI-RGSM

The major advantage of PI-RGSM is the self-supervised learning scheme. To verify whether the hard constraint method enables PI-RGSM to achieve self-supervised learning and reduce the dependence on observations, experiments were conducted between PI-RGSM and baseline PINNs and their extended models. The experiment did not use observations but instead relied solely on the physics-informed loss function for model training. Since the baseline PINNs and their extended models cannot modify the predefined hydrogeological parameters after training, the experiment was set to fixed conditions: precipitation $W$=0.004$m/d$, homogeneous hydraulic conductivity $K$=0.7$m/d$, and canal heights of $2m$. The prediction results were compared with reference data generated by MODFLOW. For the baseline PINNs, the loss term weights were set as $\lambda_{PDE} = 100$, $\lambda_{BC} = 1$, $\lambda_{IC} = 1$, $\lambda_{EC} = 1$, and $\lambda_{OB} = 1$ based on empirical experience to balance their contributions to the overall loss function. The network structure of baseline PINNs was determined through hyperparameter tuning, with 7 layers and 50 neurons per layer identified as the optimal configuration. Detailed tuning processes and results are provided in the appendix F.

The performance metrics of models are calculated and shown in Table 2. Results show that the baseline PINNs fail to capture the variation trend of phreatic head, with high errors. Extensions like RCNN and GW-PINN demonstrate significantly better performance compared to the baseline, achieving $R^2$ values of 0.95862 and 0.96246, respectively, but they are limited to a single hydrogeological environment. However, PI-RGSM further outperforms them under the same test conditions, achieving an $R^2$ accuracy exceeding 0.98. This result demonstrates that the PI-RGSM successfully eliminates dependence on observations and realizes self-supervised learning.

Table 2: Performance of baseline PINNs and PI-RGSM without observations

|  | *MAE* | *RMSE* | $R^2$ |
|---|---|---|---|
| Baseline PINNs (Raissi et al., 2019) | 0.33046 | 0.36150 | -4.32762 |
| RCNN (Sun et al., 2023) | 0.13026 | 0.11265 | 0.95862 |
| GW-PINN (Zhang et al., 2022) | 0.12658 | 0.09523 | 0.96246 |
| PI-RGSM | 0.01391 | 0.01651 | 0.98889 |

Another advantage of the PI-RGSM is its adaptivity to different source/sink terms and hydrogeological parameters for inference without retraining. To verify whether PI-RGSM can accurately capture phreatic surface based on source/sink term and hydrogeological parameter without retraining, the trained PI-RGSM is applied to different source/sink terms and hydrogeological parameters as inputs. The results are compared with reference data generated by MODFLOW.

(1) **The influence of precipitation and hydraulic conductivity:** To evaluate the performance of the PI-RGSM under different source/sink terms and hydrogeological conditions, a series of experiments was designed to simulate different groundwater seepage scenarios by adjusting precipitation $W$ and hydraulic conductivity $K$. All experiments were carried out under fixed left and right canal water levels $H_a$=$H_b$=$2m$ to ensure consistent boundary conditions. Various combinations of hydrogeological parameters were set up in the experiment. First, the effect of source/sink terms (precipitation $W$) on the model was tested under a fixed hydraulic conductivity of $K$=0.700$m/d$ with three differ-

ent precipitation levels: low, medium, and high ($W$=0.002$m/d$,$W$=0.004$m/d$, and $W$=0.006$m/d$). Then, the hydraulic conductivity was adjusted, and three different values were tested ($K$=0.500$m/d$, $K$=0.700$m/d$, and $K$=0.900$m/d$) under a constant precipitation rate of $W$=0.005$m/d$.

The test results are shown in Table 3. For each set of test configurations, the hydraulic head predicted by the model is displayed as a 3D graph (Figure 3 and 4), with the phreatic line profile extracted at $y$=5$m$ for comparison with the reference data generated by MODFLOW. The 3D image on the top displays the prediction of the phreatic surface, while the profile on the bottom shows the phreatic line at $y$=5$m$; The blue curve represents the model prediction, and the red curve denotes the reference curve from MODFLOW. The highest hydraulic head obtained from the prediction curve and the reference curve are marked in the figure. The results demonstrate that the phreatic surface predicted by the model is smooth, consistent with the typical morphology of groundwater seepage in river-canal system, and almost completely matches the reference curve from MODFLOW. In Figure 3, as precipitation increases from 0.002$m/d$ to 0.006$m/d$, the model predicts a corresponding rise in hydraulic head. For example, at the lowest precipitation (0.002$m/d$), the predicted maximum hydraulic head is approximately 2.29$m$, while at the highest precipitation (0.006$m/d$), it rises to about 2.75$m$, demonstrating the model's sensitivity and adaptability to changes in precipitation. In the case of different hydraulic conductivity (Figure 4), PI-RGSM also shows a high degree of accuracy, with a mean $R^2$ exceeding 0.977 relative to the reference data. These experiments confirm the high adaptability and accuracy of the model under different source/sink terms and hydrogeological conditions and its physical consistency and reliability in capturing groundwater seepage behavior.

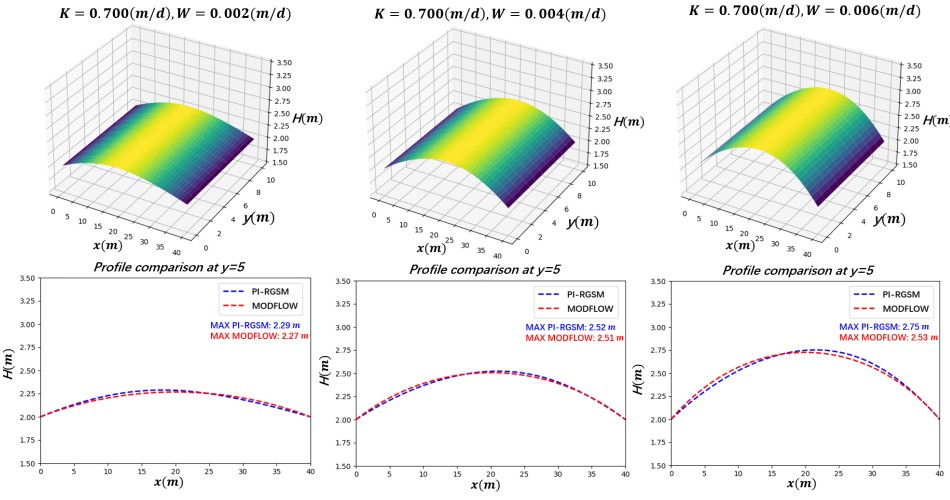

Figure 3: Phreatic surfaces from PI-RGSM under different precipitation conditions.

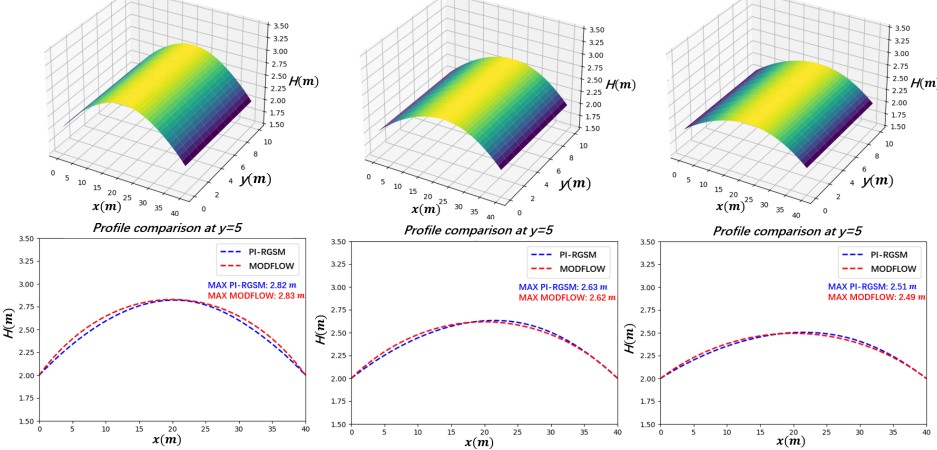

Figure 4: Phreatic surfaces from PI-RGSM under different hydraulic conductivities.

(2) **The influence of boundary condition changes:** A series of experiments were designed to evaluate the response of the PI-RGSM to changes in boundary conditions, maintaining precipitation $W$ and hydraulic conductivity $K$ at 0.004$m/d$ and 0.700$m/d$, respectively, to ensure consistency in

hydrogeological parameters. The key variables in this experiment are the water levels of the left and right canals (i.e., $H_a$ and $H_b$). Here, $H_b$ is kept constant at $2.000m$, while $H_a$ varied at $2.500m$, $3.000m$, and $3.500m$. This configuration aims to simulate diverse water level differences to examine the model performance under varying boundary conditions.

The test results are summarized in Table 4. To visualize the prediction performance of the model under different boundary water levels, 3D graphs are plotted. As $H_a$ gradually increases from $2.500m$ to $3.500m$, the predicted hydraulic head changes accordingly (shown in Figure 5). In particular, the profile comparison at $y=5m$ reveals that the predicted hydraulic head gradually increases with the elevation of $H_a$, maintaining a high degree of consistency with the MODFLOW reference. These results demonstrate the response of the PI-RGSM under changing boundary conditions and emphasizes its high prediction accuracy, with an mean $R^2$ exceeding 0.99.

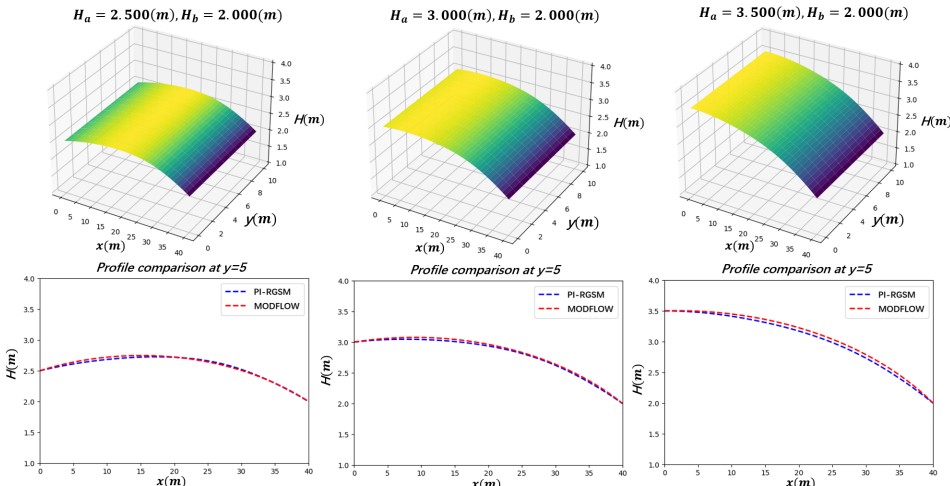

Figure 5: Phreatic surfaces from PI-RGSM under different boundary conditions.

Table 3: Performance of PI-RGSM under different source/sink terms and hydrogeological

| Input features | | MAE | RMSE | $R^2$ |
|---|---|---|---|---|
| $H_a = H_b = 2.000$ | | | | |
| | $W = 0.002$ | 0.01621 | 0.01847 | 0.95065 |
| $K = 0.700$ | $W = 0.004$ | 0.01391 | 0.01651 | 0.98889 |
| | $W = 0.006$ | 0.02639 | 0.03064 | 0.98110 |
| | $K = 0.500$ | 0.03388 | 0.03934 | 0.97598 |
| $W = 0.003$ | $K = 0.700$ | 0.02013 | 0.02394 | 0.98420 |
| | $K = 0.900$ | 0.01685 | 0.01938 | 0.98390 |

Table 4: Performance of PI-RGSM under different boundary conditions

| Input features | MAE | RMSE | $R^2$ |
|---|---|---|---|
| $W = 0.004, K = 0.700$ | | | |
| $H_a = 2.500, H_b = 2.000$ | 0.02189 | 0.02392 | 0.99410 |
| $H_a = 3.000, H_b = 2.000$ | 0.01701 | 0.02077 | 0.98926 |
| $H_a = 3.500, H_b = 2.000$ | 0.04012 | 0.04411 | 0.99002 |

## 4.2 PI-RGSM-K

To evaluate the performance of the PI-RGSM-K in handling heterogeneous hydraulic conductivity fields, experiments was conducted by under different conditions of source/sink terms and water levels of left and right canals. Under the complex geological conditions of heterogeneous hydraulic conductivity field, the performance metrics were calculated and shown in Table 5. The predicted phreatic surface was visualized using 3D graphs and cross-sectional diagrams of the phreatic line (Figure 6 and 8). Vertical and dashed lines indicated the positions of the predicted and referenced maximum hydraulic heads.

In the experiment different precipitation (Figure 6), the phreatic surface predicted by PI-RGSM-K rose accordingly as precipitation increased from $0.003m/d$ to $0.007m/d$, especially under high precipitation conditions. This demonstrates that PI-RGSM-K can accurately capture the impact of

precipitation change on the phreatic surface in heterogeneous hydraulic conductivity field. The predicted results are highly consistent with the reference curve of MODFLOW, with $R^2$ exceeding 0.99. In the experiment in response to boundary condition changes (results in appendix G), the hydraulic head predicted by the model changes correspondingly as the water level $H_a$ increases from $2.500m$ to $3.500m$. In the profile comparison at $y$=5$m$, it is evident that the predicted hydraulic head increases with the elevation of $H_a$. The results indicate a high degree of consistency with the MODFLOW reference, with a mean $R^2$ above 0.98. It is further proven that the PI-RGSM-K has strong adaptability and accuracy in processing boundary conditions, especially under heterogeneous hydraulic conductivity field, and can accurately capture changes in phreatic surface morphology caused by river-canal water levels changes. Additionally, when responding to changes in precipitation and boundary conditions, it can be observed that the groundwater flow increases on the left side as the hydraulic conductivity field gradually decreases from the left side to the right side. This result in the shift of the peak value of the phreatic surface to the right side, as clearly seen in the figures. This high level of physical consistency proves its superior performance in groundwater seepage prediction tasks involving heterogeneous hydraulic conductivity field. The performance remains stable under various hydrogeological conditions, which demonstrate the potential application of the model in future groundwater resource management and hydrology research. Future studies will further explore the applicability and optimization methods of the model in a wider range of groundwater seepage scenarios.

Table 5: Performance of PI-RGSM under different precipitations and boundary conditions

| *Input features* | | $MAE$ | $RMSE$ | $R^2$ |
|---|---|---|---|---|
| | $W = 0.003$ | 0.01037 | 0.01335 | 0.99113 |
| $H_a = 2.000, H_b = 2.000$ | $W = 0.005$ | 0.01419 | 0.01650 | 0.99446 |
| | $W = 0.007$ | 0.01514 | 0.01758 | 0.99641 |
| | $H_a = 2.500, H_b = 2.000$ | 0.02778 | 0.02814 | 0.98821 |
| $W = 0.003$ | $H_a = 2.700, H_b = 2.000$ | 0.03245 | 0.03308 | 0.98057 |
| | $H_a = 3.000, H_b = 2.000$ | 0.02157 | 0.02501 | 0.98149 |

Figure 6: Phreatic surfaces from PI-RGSM-K under different precipitation conditions.

## 5 CONCLUSION

This study proposes the PI-RGSM and its extension PI-RGSM-K based on PINNs which significantly improve the accuracy and adaptability of groundwater seepage prediction. The main contributions are as follows: **(1)** Self-supervised learning of PI-RGSM is achieved by integrating hard constraints of boundary and initial conditions, reducing the dependence on observations. **(2)** PI-RGSM demonstrates strong generalization capabilities, as it can adapt to varying hydrogeological conditions without requiring retraining, significantly improving its practicality in real-world applications. **(3)** PI-RGSM-K is proposed to simulate heterogeneous porous media by considering heterogeneous hydraulic conductivity. This study demonstrates that integrating physical knowledge into a neural network framework provides an effective scientific method for the sustainable management and protection of groundwater resources, highlighting the great potential and practical value of the physics-informed method in groundwater seepage modeling.

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

## A  RELATED WORKS

With the evolution of scientific paradigms (Caíno-Lores et al., 2020), researches on natural science has mitigated from empirical and theoretical approaches to computational and data-intensive scientific paradigms driven by computer simulations and big data (Babovi & Bajat, 2023; Li et al., 2023). In the field of computational fluid dynamics, numerical models including FLUENT (Galeev et al., 2013), MODFLOW (Brunner et al., 2010; Harbaugh, 2005; Hariharan & Shankar, 2017), and COMSOL Multiphysics (Multiphysics, 1998), have achieved accurate simulations of physical processes like complex multiphase flow, solute transport, and interactions between surface and soil water within groundwater systems. Important issues in those spatial discretization methods for solving the partial differential equations (PDEs) of groundwater are the computationally intensive. Moreover, the implementation of these models is typically challenging for personnel at water resource management agencies, as highly specialized knowledge and extensive experience in numerical modeling are required. Continuous advancements in data-intensive scientific paradigms have empowered deep neural networks to achieve significant breakthroughs in image recognition (Krizhevsky et al., 2017), meteorological modeling (Wan et al., 2018), material informatics (Niu et al., 2020), hydrology (Maxwell et al., 2021), and mathematics(Choudhary et al., 2020), due to the powerful feature extraction (Yu et al., 2013; Hinton & Salakhutdinov, 2006) and complex function approximation abilities (Cybenko, 1989). Nonetheless, as "black box" models, neural networks exhibit a lack of transparency in their decision-making processes and depend significantly on extensive training data, limiting their use in groundwater research. Recently, by combining physical laws with neural networks, the physics-informed neural network (PINNs) method has improved the interpretability of models and diminished reliance on extensive datasets, providing an efficient and accurate approach for groundwater system modeling, and thus promoting the scientific and reliability of groundwater prediction and management decisions.

The concept of PINNs was first proposed by Raissi et al. (2019). In their approach, the neural network was trained by constructing physical constraints, and optimized by using the residuals based on initial conditions, boundary conditions, and governing equations as loss functions. Neural network derivatives were computed through automatic differentiation (AD). This method effectively solves both continuous and discrete PDEs, marking a new era of integration of physical knowledge and deep learning. The introduction of PINNs has significantly expanded the scope of neural networks in scientific research and engineering applications, while markedly enhancing the prediction accuracy, generalization ability, and interpretability of models. Since then, researchers have continued to explore and innovate within the PINNs framework, applying it to various fields such as fluid mechanics (Haghighat et al., 2021), structural dynamics (Moradi et al., 2023), acoustics (Song et al., 2021), and solid mechanics (Wang et al., 2023). In hydrological research, PINNs offer a new perspective for more convenient and efficient prediction of groundwater flow with their unique physical guidance characteristics. Wang et al. (2020) proposed a theoretically guided neural network model (TgNN) based on the loss function of groundwater seepage differential equations,

which improved the generalization performance of the groundwater seepage model with limited observational data. Additionally, the team demonstrated that the physics-informed model could effectively use physical information to predict groundwater seepage responses beyond the training set through groundwater seepage transfer learning tasks. Cho & Kim (2022) used an LSTM network to regress the residual differences between a neural network model and the Weather Research and Forecasting model-Hydro (WRF-Hydro). They developed a hybrid model (WRF-Hydro-LSTM) to predict groundwater levels, and experimental results indicated that this hybrid model achieved higher prediction accuracy, less sensitivity to training datasets, and better generalization performance than the LSTM network or WRF-Hydro model alone. Wang et al. (2021a) proposed a neural network constrained by geostatistical information and a physics-guided autoencoder based on convolutional neural networks (CNNs) (Wang et al., 2021b), applying the physical guidance method to the parameter inversion of groundwater hydraulic conductivity fields. The model's accuracy and practicality were further enhanced. The study by (Pashaei Kalajahi et al., 2022) demonstrated that, even with limited data, physics-guided neural networks could accurately estimate key parameters such as hydraulic conductivity and porosity of the medium, showcasing their strong ability to combine data-driven approaches with physical guidance. Daolun et al. (2021) established a new specialized neuron model incorporating pressure gradient information and proposed an algorithm called signpost neural network (SNN), which significantly improves the accuracy of solving unsteady seepage partial differential equations. Although significant progress has been made in improving groundwater seepage models using PINNs, current models still heavily depend on the adequacy and quality of observed data and remain sensitive to outliers. If the training datasets are insufficient or contain corrupted data, the simulation results may exhibit considerable bias. Moreover, after the model has been trained, its applicability is generally limited to specific hydrogeological parameter settings, making it difficult to generalize to broader or unforeseen hydrogeological scenarios, thus restricting the model's versatility and practicality.

## B  BASIC LAW OF GROUNDWATER SEEPAGE

Without considering changes in water density, the flow of groundwater in a three-dimensional aquifer within porous media can be expressed by the following partial differential equation:

$$\frac{\partial}{\partial x}\left(K_{xx}\frac{\partial H}{\partial x}\right) + \frac{\partial}{\partial y}\left(K_{yy}\frac{\partial H}{\partial y}\right) + \frac{\partial}{\partial z}\left(K_{zz}\frac{\partial H}{\partial z}\right) + W = \mu_s M \frac{\partial H}{\partial t} \tag{15}$$

Where $t$ is time ($T$); $K_{xx}$, $K_{yy}$, and $K_{zz}$ are the hydraulic conductivities ($LT^{-1}$) along the $x$, $y$, and $z$ axes, respectively. $H$ is the hydraulic head ($L$) at the corresponding space-time point; $W$ is the source/sink term of groundwater (such as precipitation, pumping, etc.); $\mu_s$ is the specific storage ($L^{-1}$) of the aquifer below the free surface. $M$ is the thickness of the aquifer. Combined with fixed solution conditions (boundary conditions and initial conditions), it can reflect the water balance relationship per unit volume and per unit time under Darcy flow conditions.

## C  DEEP NEURAL NETWORK

The Deep Neural Network (DNN) consists of an input layer, several hidden layers, and an output layer, where each layer contains multiple neurons connected by a weight matrix (which may include bias terms). Consider a deep neural network with $L$ layers, denoted as $Y = f_{net}(X, \theta)$, with $n^l$ neurons in the $L$-th layer. The number of neurons in the input layer equals the dimension of the input feature vector $X$, and the number of neurons in the output layer equals the dimensionality of the output vector $Y$. The value of the $i$-th neuron in $L$-th layer, $z_{l,i}$, is computed as the product of the weight $W_{l,i}^T$ and neurons from the previous layer, plus the bias term $b_{l,i}$:

$$z_{l,i} = W_{l,i}^T a_{l-1} + b_{l,i} \tag{16}$$

The output of neurons is nonlinearly transformed by the activation function, which provides nonlinear capability to the network. After processing $z_{l,i}$ with the activation function $\sigma$, the output of the $i$-th neuron in the $L$-th layer, $a_{l,i}$, Lis obtained:

$$a_{l,i} = \sigma(z_{l,i}) \tag{17}$$

The final output of the network is produced by the output layer of the multi-layer neural network, and the network training process relies on a loss function that quantitatively assesses the difference between the network's predictions and the actual target values.

## D  PERFORMANCE METRICS

To evaluate the performance of the model, three main indicators were used for quantitative analysis: Mean Absolute Error ($MAE$), Root Mean Square Error ($RMSE$), and Determination Coefficient ($R^2$). Their definitions are as follows:

$$MAE = \frac{1}{n} \sum_{i=1}^{n} |y_i - \hat{y}_i| \tag{18}$$

$$RMSE = \sqrt{\frac{1}{n} \sum_{i=1}^{n} (y_i - \hat{y}_i)^2} \tag{19}$$

$$R^2 = 1 - \frac{\sum_{i=1}^{n} (y_i - \hat{y}_i)^2}{\sum_{i=1}^{n} (y_i - \bar{y})^2} \tag{20}$$

Where n represents the total number of samples, $y_i$ represents the reference output from MODFLOW of the $i$-$th$ sample, $\hat{y}_i$ represents the predicted value of the $i$-$th$ sample by the DNN model, and $\bar{y}_i$ is the average of the reference values of all samples. Ideally, the closer $MAE$ and $RMSE$ are to 0, the smaller the prediction error and the higher the accuracy of the model. The closer the $R^2$ value is to 1, the better prediction the DNN model achieves.

## E  DETAILS OF PI-RGSM-K MODEL

Given that the PI-RGSM-K model is a specific application of the PI-RGSM model under conditions of a heterogeneous hydraulic conductivity field, and since the structures of both models are similar, most of the experimental settings are identical to PI-RGSM as follows:

(1) As the hydraulic conductivity field has been integrated into the physics-informed loss function, the model input features are spatial-temporal coordinates $(x, y, t)$, canal water levels $(H_a, H_b)$, and source/sink term $W$. The model still uses a random method to generate input data for self-supervised training (shown in Table 6).

(2) A heterogeneous hydraulic conductivity field is introduced, with $K$ defined as a function that varies linearly with $x$ (i.e., $K = -0.01x + 0.8$). The hydraulic conductivity linearly changes from $0.8m/d$ at the left boundary to $0.4m/d$ at the right boundary.

Table 6: Input features and value ranges of the PI-RGSM-K

| Input features | $x(m)$ | $y(m)$ | $t(d)$ | $W(m/d)$ | $H_a(m)$ | $H_b(m)$ |
|---|---|---|---|---|---|---|
| Value ranges | [0,40] | [0,10] | [0,100] | [0,0.007] | [2,3.5] | [2,3.5] |

## F  HYPERPARAMETER TUNING PROCEDURE OF BASELINE PINNs

The hyperparameter settings of the neural network, including the learning rate, layers, and neurons, directly influence the predictive accuracy of baseline PINNs. Figure 7 illustrates the effects of these factors on the $L^2$ relative error. As the initial learning rate decreases, the error drops significantly and stabilizes below 1%, reaching a minimum at a learning rate of 0.001. However, further reducing the learning rate introduces additional errors. The results show that increasing the number of layers generally improves prediction accuracy, but the improvement becomes negligible beyond 5 layers. Similarly, increasing the number of neurons per layer reduces the error, with the most significant reduction observed up to 30 neurons per layer. After this point, further increases in neurons yield diminishing returns.

The optimal network structure identified in this study consists of 7 layers with 50 neurons per layer, achieving an $L^2$ error of 2.83%. While deeper or wider networks may further improve accuracy slightly, the additional computational cost outweighs the marginal benefits due to the relatively low

complexity of the governing equations in this study. In summary, Too many or too few neurons in hidden layers can cause overfitting or underfitting. Excessive neurons may lead to overfitting due to insufficient training data, while also increasing training time without achieving better performance. Selecting an appropriate number of neurons is essential for balancing accuracy and efficiency.

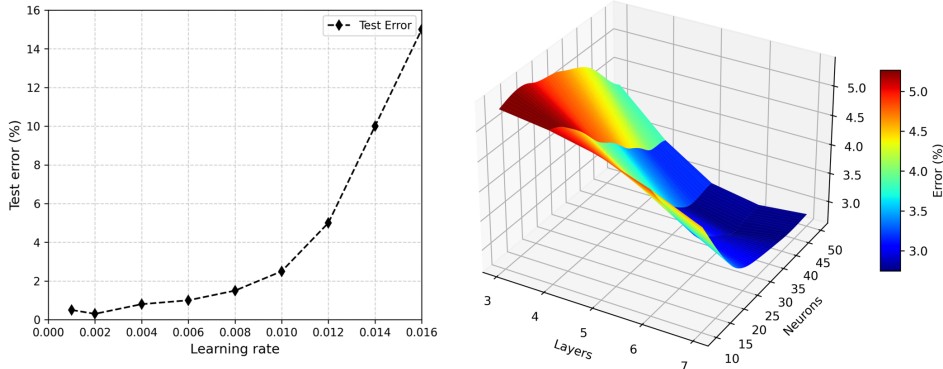

Figure 7: Impact of learning rate (left) and network parameters (Layers and Neurons) on the performance of baseline PINNs.

## G PHREATIC SURFACES FROM PI-RGSM-K UNDER DIFFERENT BOUNDARY CONDITIONS.

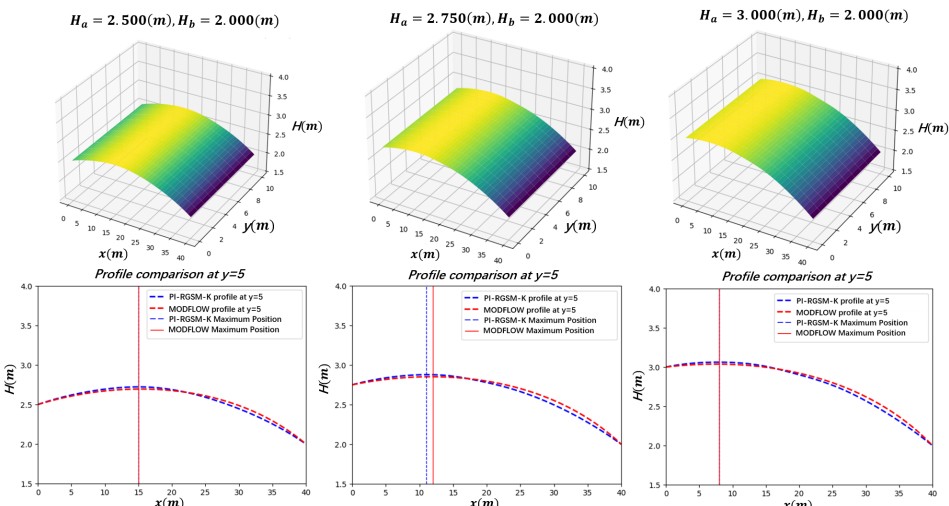

Figure 8: Phreatic surfaces from PI-RGSM-K under different boundary conditions.

