# OpenReview forum: "Groundwater Seepage Modeling in a River-Canal System based on Physics-Informed Neural Networks"
_ICLR.cc/2025/Conference — ICLR 2025 Conference Withdrawn Submission_

### Official Review · Reviewer_qtcb · 2024-10-31

**Soundness:** 1
**Presentation:** 1
**Contribution:** 1
**Rating:** 3
**Confidence:** 2

**Summary:**

This manuscript introduces two PINN-like models for predicting groundwater seepage, a task in hydrogeology.

**Strengths:**

(significance) automating the analysis of groundwater dynamics seems to be a useful application for machine learning.

**Weaknesses:**

- (clarity/novelty) The authors claim they introduce self-supervised PINNs, but I fail to comprehend how these models are more self-supervised than the original PINNs.
   The proposed models seem to use have an additional scaling and shift on top of the additional inputs compared to the baseline PINN, but I do not see how this makes these models self-supervised.
 - (novelty) This model fails to cite any relevant related work in the main paper.
   I noticed there is a related work section in the appendix, but that does not suffice.
   During reading I had the impression this is the only paper applying deep learning in the field of hydrogeology, which is obviously not the case.
   Furthermore, the main paper should discuss how the proposed method compares to existing methods.
 - (quality/clarity) I could not find information on how the baseline PINN was trained.
   E.g. section 2.2 mentions the different hyper-parameters in the loss function, but I could not find any information on what hyper-parameters were used or how they were found.
 - (quality) Each model is compared to a single baseline PINN.
   I strongly doubt that there are no stronger baselines to compare against (e.g. regular networks, non-ML methods, ...).
   Furthermore, the performance of the baseline PINN is so poor that it seems as if something went wrong during training (assuming a random model would achieve an R^2 of 0).
   Also, there are no error bars to indicate the consistency of the compared model(s).
 - (significance/novelty) I fail to find a technical contribution that would be of interest to the machine learning community.
   The PINN modifications are ad-hoc solutions for the equations at hand, and I can't imagine that this general idea has not been applied before.
   For an example of how physical constraints have been embedded in the architecture in the context of hydrology, I can refer to (Hoedt et al., 2021), but there are probably many more.

###### References
 - Hoedt et al. (2021). MC-LSTM: Mass-Conserving LSTM. Proceedings of the 38th International Conference on Machine Learning, 139, 4275–4286. http://proceedings.mlr.press/v139/hoedt21a.html

**Questions:**

1. Why are the proposed models considered self-supervised variants of PINNs?
2. Are there no other possible baselines or models to compare against?
3. How were the hyper-parameters for the baseline PINN tuned?
4. Wouldn't this paper be more interesting to hydrogeologists than to the ML community?

---

> ### Author Response · Authors · 2024-11-26
> **Rebuttal by Authors**
>
> * **Response to Question 1:**
> Thank you for your question regarding the classification of our proposed models as self-supervised variants of PINNs.  The key reason is that our models do not rely on labeled or observational data for training.  Instead, the training process is driven entirely by the governing equations (PDEs), boundary conditions, and additional hard constraints, which serve as the self-supervision signals. We have provided a more detailed explanation regarding this aspect in lines 210–215 of the manuscript to address your concerns.
> * **Response to Question 2:**
> Thank you for your feedback on the evaluation. We agree that comparing PI-RGSM with additional baselines would strengthen the study.  We considered existing extensions of PINN that are designed for solving groundwater seepage equations[1] [2]. These methods, however, are typically limited to fixed conditions such as single source/sink terms and homogeneous geological settings. When these conditions change, the networks must be retrained, which limits their practicality. To reflect this limitation, we set the experiment under fixed conditions  (precipitation 𝑊=0.004𝑚/𝑑 , homogeneous hydraulic conductivity 𝐾=0.7𝑚/𝑑, and canal heights of 2𝑚).
> We will revise the manuscript to clarify this rationale and improve the discussion of relevant baselines.
> [1] ZHANG X, ZHU Y, WANG J, et al. GW-PINN: A deep learning algorithm for solving groundwater flow equations[J/OL]. Advances in Water Resources, 2022, 165: 104243. DOI:10.1016/j.advwatres.2022.104243.
> [2] SUN J, LI X, YANG Q, et al. Hydrodynamic numerical simulations based on residual cooperative neural network[J/OL]. Advances in Water Resources, 2023, 180: 104523. DOI:10.1016/j.advwatres.2023.104523.
> * **Response to Question 3:**
> Thank you for highlighting the need for greater clarity regarding the training of the baseline PINNs and the choice of hyperparameters. The weights for different loss terms in the baseline PINNs were determined through extensive empirical adjustments to balance contributions from PDE residuals, boundary conditions, and initial conditions. We have provided the specific hyperparameter settings in the baseline PINNs section for reproducibility. Additionally, a detailed discussion of the hyperparameter tuning process, including the effects of varying the learning rate and network structure, can be found in Appendix F. This ensures a thorough explanation of the training process and the rationale behind our choices.
> * **Response to Question 4:**
> Thank you for your question. While our work addresses a specific application in groundwater seepage modeling, the contributions are not limited to hydrogeology. The proposed method introduces innovations in extending the capabilities of PINNs to handle diverse physical scenarios without retraining. This is achieved through the integration of feature fusion (e.g., incorporating hydraulic conductivity, source/sink terms, and boundary conditions as inputs) and the use of hard constraints to enforce physical laws. These techniques are broadly applicable to other PDE-driven problems in science and engineering, making the work relevant to the machine learning community interested in physics-informed methods and neural operator design.

---

> > ### Comment · Reviewer_qtcb · 2024-11-26
> >
> > I am afraid that this rebuttal does not sufficiently address my concerns. Most notably, similar to other reviewers, I still fail to understand the technical novelty of this work. Therefore, I am not inclined to alter my score.

---

### Official Review · Reviewer_CiS8 · 2024-11-03

**Soundness:** 2
**Presentation:** 1
**Contribution:** 2
**Rating:** 3
**Confidence:** 3

**Summary:**

The paper proposes two models, PI-RGSM and PI-RGSM-K, for groundwater seepage prediction using physics-informed neural networks (PINNs). These models integrate physical constraints into neural networks to enhance prediction accuracy in groundwater flow, reducing dependency on observational data and adapting well to complex seepage conditions. PI-RGSM-K extends the base model by incorporating heterogeneous hydraulic conductivity fields, showing improved adaptability in complex conditions.

**Strengths:**

1. The paper effectively applies physics-informed neural networks to model groundwater seepage, enhancing model interpretability and physical consistency.
2. By integrating hard constraints, the models reduce reliance on labeled data, making them suitable for scenarios with limited observational data.
3. The models achieve promising predictive performance.

**Weaknesses:**

1. The paper is poorly organized and lacks clarity, with essential content relegated to the appendix, leaving the main text insufficiently self-contained. See questions below for specific issues.
2. Although applying PINNs to groundwater seepage is relatively novel within the specific application area, the paper contributes little in terms of new machine learning techniques.
3. The model relies on manual tuning of hyperparameters (e.g., loss weights, threshold $H_{EC}$), but the paper lacks experiments analyzing the impact of these parameters.
4. The paper fails to compare the proposed method with existing machine learning models for groundwater seepage.

**Questions:**

1. "However, the ”black box” nature of DNN exhibiting a lack of transparency in their decision-making processes and the significant dependence on extensive training data, limit their use in groundwater research." These are general drawbacks of DNN, are there any unique challenges to groundwater modeling?
2. "Although significant progress has been made in improving groundwater seepage models using PINNs, current models still heavily depend on the adequacy and quality of observed data and remain sensitive to outliers." Are there references supporting this? If these methods have known limitations, why not compare them experimentally?
3. It's better to define $\mu$ and $t$ explicitly in Section 2.1 for clarity.
4. The study uses randomly generated hydraulic conductivities and source/sink terms to enrich the training data. Is random generation commonly accepted in groundwater modeling, and does it effectively capture real-world scenarios?
5. Do $\phi$ and $\varphi$ represent the same meaning in this paper? Their mixed usage leads to confusion.
6. $H_{EC}$ is not clearly defined, nor is its role in the model explained.
7. Why use $RES$ in section 2 while use $LOSS$ in section 3?
8. Section 4.1 states, "The experiment did not use observations," which conflicts with Section 2.2(4), which introduces observed data constraints. In addition, Section 2.2 is confusing since it introduces multiple terms while it looks like most of them ($RES_{BC}, RES_{IC}, RES_{OC}$) are not used in the experiments.
9. Is there a specific reason for choosing $K=-0.01x+0.8$?

Some minor suggestions:
1. Use proper LaTeX notation for quotes: \`\`example text'' for left and right quotes.
2. Figures 2 and 3 appear nearly identical. Consider combining them into a single figure that highlights the distinctions between PI-RGSM and PI-RGSM-K for conciseness.

---

> ### Author Response · Authors · 2024-11-26
> **Rebuttal by Authors**
>
> * **Response to Weaknesses 2:**
> Thank you for your feedback regarding the technical contribution of our work. While we acknowledge that embedding physical constraints into neural networks has been explored in various contexts, our key contribution lies in addressing the limitations of the original PINN when applied to groundwater seepage modeling. Specifically, standard PINNs are designed to solve single-scenario problems and lack generalization ability. For example, solving seepage problems with different source/sink terms or geological conditions typically requires retraining the neural network from scratch. In contrast, our proposed approach incorporates feature fusion, where permeability coefficients, source/sink terms, boundary conditions, and spatiotemporal coordinates are used as inputs. This allows our model to generalize across diverse scenarios without retraining, significantly enhancing its efficiency and adaptability.
> * **Response to Weaknesses 3:**
> While it is true that PI-RGSM relies on manual tuning of hyperparameters such as loss weights and the threshold $H_{EC}$, the proposed method significantly reduces the number of hyperparameters compared to the baseline PINN by embedding hard constraints. This eliminates the need to tune additional loss terms for boundary and initial conditions (e.g., $RES_{BC}$ and $RES_{IC}$ ), which are required in baseline PINN models.
> To address this further, we have added a detailed description of the hyperparameter tuning process for the baseline PINNs in the appendix F. This addition highlights the complexity and effort required for tuning in the baseline model compared to PI-RGSM, which simplifies this process. We believe this distinction underscores the practical advantage of PI-RGSM.
> * **Response to Weaknesses 4:**
> Thank you for your comment. In response, we have updated the manuscript to include a comparison between the proposed method and existing machine learning models for groundwater seepage, particularly extensions of PINN such as RCNN and GW-PINN. This comparison is summarized in a newly added table, which highlights the performance metrics of these models alongside PI-RGSM. The results demonstrate the advantages of PI-RGSM in terms of accuracy and its ability to generalize across different hydrogeological conditions without retraining.
> ----
> * **Response to Question 1:**
> Thank you for your insightful question. In addition to the general drawbacks of deep neural networks (DNNs), the primary challenge in groundwater modeling is the scarcity of labeled data. Observational data in groundwater systems are often sparse, expensive to collect, and unevenly distributed, which limits the feasibility of purely data-driven models. Our approach specifically addresses this issue by leveraging physics-informed modeling and hard constraints to reduce the reliance on labeled data while maintaining physically consistent predictions.
> * **Response to Question 2:**
> Thank you for your insightful comment. The comparison with the baseline PINN in our experiments was primarily designed to validate the self-supervised nature of the proposed PI-RGSM model, specifically its ability to perform well without using labeled data. The poor performance of the baseline PINN highlights its limitations in this data-scarce setting, further emphasizing the advantages of PI-RGSM. To address your suggestion, we have included additional stronger baselines (e.g., RCNN and GW-PINN) in our comparisons to provide a more comprehensive evaluation of PI-RGSM.
> * **Response to Question 3:**
> Thank you for your suggestion. We have explicitly defined \mu and t in Section 2.1 to improve clarity. Specifically, $\mu$ represents the hydraulic conductivity, and $t$ denotes time.
> * **Response to Question 4:**
> Thank you for your question. Due to the scarcity of real geological data in groundwater systems, we used randomly synthesized data to enrich the training dataset. These randomly generated datasets have been applied and validated in professional groundwater simulation software such as MODFLOW, ensuring their relevance to practical scenarios. This approach enables us to simulate diverse geological conditions, including variability in hydraulic conductivity and source/sink terms, while addressing the limitations posed by the lack of real-world data. We will clarify this point in lines 307-312 of the manuscript.

---

> ### Author Response · Authors · 2024-11-26
> **Rebuttal 2  by Authors**
>
> * **Response to Question 5:**
> Thank you for pointing this out. We confirm that $\phi$ and $\varphi$ represent the same meaning in the paper. To avoid confusion, we have revised the manuscript to use a consistent notation, $φ$, throughout the text.
> * **Response to Question 6:**
> Thank you for your comment. The definition of $H_{EC}$ and its role in the model are provided in lines 162–166 of the manuscript.
> * **Response to Question 7:**
> Thank you for your observation. In Section 2, the construction of the loss function is based on the residuals of the PDE, initial conditions, and boundary conditions, so we use $RES$ to represent these residuals. In Section 3, the focus shifts to the optimization process of the model, where these residuals are aggregated and treated as a total loss function, hence the use of $LOSS$.
> * **Response to Question 8:**
> Thank you for pointing out the potential confusion regarding the use of observed data and the loss terms introduced in Section 2.2. We would like to clarify that Section 2.2 describes the loss function construction for the baseline PINNs, which includes terms such as $RES_{BC}$, $RES_{IC}$, and $RES_{OC}$. These terms are specifically used to train the baseline PINNs and are included to provide a comprehensive explanation of the baseline method. In contrast, our proposed model, which builds upon and improves the baseline PINN, does not rely on observed data constraints ($RES_{OC}$) during training. This distinction had be made clearer in the revised manuscript to avoid any ambiguity.
> * **Response to Question 9:**
> Thank you for your question. The choice of $K=-0.01x+0.8$ is intended as a representative example, aiming to parameterize the hydraulic conductivity $K$ as a function of spatial coordinates $x$ and $y$, making it more reflective of realistic geological scenarios. This approach demonstrates how $K$ can be incorporated into the model as a spatially varying parameter. However, $K$ can also be parameterized using other functions of $x$ and $y$, or even as a randomly generated hydraulic conductivity field. When $K$ is a random permeability field that can better reflect the real geological environment, we can build a separate neural network to take $(x,y)$ as the input of the neural network, and $K$ as the output. Using the powerful function approximation ability and automatic differential characteristics of the neural network, we can obtain the gradient characteristics of $K$ about $x$ and $y$, which is the direction we need to improve in the future. We  clarify this point in lines 329-337 of the manuscript.
> ----
> Thank you for your suggestion. We have combined Figures 2 and 3 into a single figure to highlight the distinctions between PI-RGSM and PI-RGSM-K for conciseness. In the updated figure 2, the neural network input components and the embedded PDE information, such as the hydraulic conductivity and source/sink term, are clearly illustrated. The blue line section represent PI-RGSM, while red dashed section indicate the additional components introduced in PI-RGSM-K, effectively emphasizing their differences.

---

> > ### Comment · Reviewer_CiS8 · 2024-11-26
> >
> > Thanks for your rebuttal. The current manuscript looks better but is still below the ICLR bar, mainly because the paper structure is still unclear and the technical novelty is limited. Therefore, I would maintain my score.

---

### Official Review · Reviewer_RrU2 · 2024-11-04

**Soundness:** 2
**Presentation:** 2
**Contribution:** 2
**Rating:** 3
**Confidence:** 4

**Summary:**

The paper introduced a physics-informed river-canal groundwater seepage model (PG-RGSM) and its variant PG-RGSM-K to model the groundwater seepage described using a PDE. Specifically, the paper proposes a way to incorporate boundary and initial conditions as hard-constraints in PINNs using a boundary constraint function. Further, the paper introduces an input-feature fusion that enables PINNs to learn solutions with different PDE coefficients.

**Strengths:**

1. The problem of predicting river-canal groundwater seepage is interesting and is very important for a number of scientific applications.
2. The proposed method of enforcing hard-constraints is interesting.

**Weaknesses:**

1. The overall evaluation of the paper is weak. The paper mentions (in lines 167-169) that the main challenges in PINNs are the gradient pathologies with multiple loss terms, and the tedious nature of adjusting the trade-off hyper-parameters between these different terms. There is a rich body of work that addresses these challenges in PINNs, and should be used as baselines for comparison. Here are some of them: [1, 2, 3, 4]
2. Lack of comparison with methods that can enforce hard boundary constraints [5] in PINNs.
3. The “Input feature fusion” concept introduced in the paper is similar to the Neural Operators [6, 7, 8], where the ML model learns a family of PDEs instead of one single instantiation of a PDE. Therefore, the proposed PG-RGSM architecture seems like a simpler variant of the DeepONet [7] architecture, where both the coefficients and the spatio-temporal inputs are combined into one model instead of having a separate branch and trunk networks. Comparing against neural operators like DeepONets would further improve the paper.

[1] Wang, S., Teng, Y., and Perdikaris, P. Understanding and mitigating gradient flow pathologies in physics-informed neural networks. SIAM Journal on Scientific Computing, 43(5):A3055–A3081, 2021.

[2] Wang, S., Yu, X., and Perdikaris, P. When and why pinns fail to train: A neural tangent kernel perspective. Journal of Computational Physics, 449:110768, 2022c.

[3] Krishnapriyan, A., Gholami, A., Zhe, S., Kirby, R., and Mahoney, M. W. Characterizing possible failure modes in physics-informed neural networks. Advances in Neural Information Processing Systems, 34, 2021.

[4] Daw, Arka, Jie Bu, Sifan Wang, Paris Perdikaris, and Anuj Karpatne. "Mitigating propagation failures in physics-informed neural networks using retain-resample-release (r3) sampling." arXiv preprint arXiv:2207.02338 (2022).

[5] Liu, Songming, Hao Zhongkai, Chengyang Ying, Hang Su, Jun Zhu, and Ze Cheng. "A unified hard-constraint framework for solving geometrically complex pdes." Advances in Neural Information Processing Systems 35 (2022): 20287-20299.

[6] Kovachki, Nikola, Zongyi Li, Burigede Liu, Kamyar Azizzadenesheli, Kaushik Bhattacharya, Andrew Stuart, and Anima Anandkumar. "Neural operator: Learning maps between function spaces with applications to pdes." Journal of Machine Learning Research 24, no. 89 (2023): 1-97.

[7] Lu, Lu, Pengzhan Jin, and George Em Karniadakis. "Deeponet: Learning nonlinear operators for identifying differential equations based on the universal approximation theorem of operators." arXiv preprint arXiv:1910.03193 (2019).

[8] Li, Zongyi, Nikola Kovachki, Kamyar Azizzadenesheli, Burigede Liu, Kaushik Bhattacharya, Andrew Stuart, and Anima Anandkumar. "Fourier neural operator for parametric partial differential equations." arXiv preprint arXiv:2010.08895 (2020).

**Questions:**

1. It might be interesting to investigate why PINNs fail for this particular PDE (lines 167-169).
2. In the Introduction section, there are a couple of references to the Appendix. Adding appropriate citations (along with ref. to the appendix) would improve the readability.
3. The Related Works section provided in the Appendix was really helpful. I would suggest shrinking it and adding it to the main paper to help readers familiarize with the current state-of-the-art.
4. The paper claims that PG-RGSM improves convergence rates (line 179). It will strengthen the proposed method if it is demonstrated empirically.
5. Eqn 11: The $t$ present in both the numerator and denominator of $C(x, y, t)$. Shouldn’t the denominator be $T$ (which is the final time)?
6. How are the Dritchlet boundary conditions obtained for this problem?

---

> ### Author Response · Authors · 2024-11-26
> **Rebuttal by Authors**
>
> * **Response to Weaknesses:**
> Thank you for your insightful comment regarding the similarity between the proposed input feature fusion concept and Neural Operators such as DeepONet. We would like to clarify that the key distinction lies in the context and assumptions of the problem. DeepONet and similar neural operator approaches are fundamentally data-driven and require a significant amount of labeled training data to learn the solution operator for a family of PDEs. In contrast, our work addresses scenarios where observational data is sparse or unavailable, which is a common challenge in groundwater modeling. Our proposed method leverages physics-based self-supervision and hard constraints to train the model without relying on labeled data.
> Given this foundational difference in methodology and problem context, we believe that a direct comparison with neural operators like DeepONet is not entirely appropriate or necessary for this study.
> ----
> * **Response to Question 1:**
> Thank you for your comments regarding the evaluation and comparison in our study. The gradient pathologies of multiple loss terms are indeed inherent challenges in PINNs, and while there is a rich body of work addressing these issues, their application to groundwater flow modeling is still relatively limited. In our revised manuscript, we have added comparisons with existing PINN-based models applied to groundwater flow modeling in the baseline section. These comparisons include models like GW-PINN and RCNN, which address the balancing of different loss terms and their contributions. By including these models, we further validate the self-supervised learning capability of PI-RGSM. While the improvements in some cases may not seem substantial, the key advantage of PI-RGSM lies in its generalization capability, as highlighted in the manuscript (second major advantage). Specifically, PI-RGSM requires only a single training session to adapt to varying groundwater flow conditions within the same geological setting. To the best of our knowledge, very few works have achieved this level of flexibility in groundwater modeling.
> * **Response to Question 2 and 3:**
> Thank you for your valuable suggestions regarding the Introduction and Related Works sections.  We have revised the Introduction to include proper citations for referenced works alongside references to the Appendix.  This will improve the clarity and readability of the manuscript.
> For Question 3, we appreciate your feedback on the usefulness of the Related Works section in the Appendix.  To enhance the flow of the paper and provide readers with better context about the state-of-the-art, we have condensed this section and incorporate it into the main manuscript while ensuring that the key points are preserved.  The revised structure will balance readability with space limitations and improve the accessibility of our work.
> * **Response to Question 4:**
> Thank you for your comment. The improvement in convergence rates for PI-RGSM is reflected through the model’s higher prediction accuracy for the phreatic head compared to baseline methods, as shown in Table 2. Instead of focusing on loss function experiments, we chose to evaluate convergence indirectly by assessing the model's ability to predict hydraulic head more accurately under the same training conditions.
> * **Response to Question 5:**
> Thank you for your observation regarding the presence of t in both the numerator and denominator of Eq. (11). We would like to clarify that the $t$ of denominator is not intended to represent $T$(the final time). Instead, it is part of the scaling factor, $N_s$, designed to balance the magnitude of the data and ensure numerical stability during computation. Specifically, $N_s$ is set as the maximum value of the numerator $\left(l^2t\right)$ to normalize the function appropriately. After applying hard constraints, the model output at boundary and initial points directly matches $\varphi(X)$, eliminating the need for separate loss functions for these conditions. For other points, the output is constrained by the constructed PDE. This approach reduces the number of loss functions, improving the model's orientation and solving efficiency.
> To address potential misunderstandings, we have provided a detailed explanation of this design choice in lines 210–215 of the manuscript. We hope this clarifies the purpose of $N_s$ and resolves your concern.
> * **Response to Question 6:**
> Thank you for your question. The Dirichlet boundary conditions for this problem are predefined based on the physical setup of the groundwater seepage scenario. Specifically, they represent the hydraulic head values at the left and right boundaries of the canal system, which are determined from typical groundwater flow assumptions and prior knowledge of the problem domain. These values are then directly incorporated into the model as boundary constraints to ensure consistency with the physical system being modeled.

---

> > ### Comment · Reviewer_RrU2 · 2024-11-26
> >
> > Thank you to the authors for their detailed responses and for clarifying some of my questions. I have also read the reviews from the other reviewers. Despite the authors' points in the rebuttal, I maintain that PINN variants should be included as baselines to demonstrate the significance of the proposed approach.
> >
> > While I appreciate the author's efforts, I do not believe that the paper meets the high standards of ICLR due to limited novelty and the weak evaluation of the proposed method. I have updated my score accordingly.

---

### Official Review · Reviewer_tGGb · 2024-11-08

**Soundness:** 2
**Presentation:** 2
**Contribution:** 2
**Rating:** 3
**Confidence:** 4

**Summary:**

The paper presents an extension of physics-informed neural networks (PINNs) for groundwater seepage modeling, termed PI-RGSM. The proposed method enforces hard constraints on the initial and boundary conditions in the outputs of the neural network. A variant of PI-RGSM is also proposed to handle heterogeneity in hydraulic conductivity field. Experiments are performed on groundwater simulations to compare PI-RGSM with the PINN baseline.

**Strengths:**

1. Applies PINNs and its variants on a new real-world problem.
2. Provides details of the model architecture and the loss functions used in proposed PI-RGSM framework.

**Weaknesses:**

1. Evaluation is not comprehensive. PI-RGSM is not compared with any baselines except PINN (and only in Table 2). There are many extensions of the basic PINN that can be considered as additional baselines, along with operator learning methods such as FNO. It will be good to include visualizations of baselines along with PI-RGSM in the remaining tables and figures of the paper apart from Table 2.
2. Dataset looks too simple. The visualizations show that the ground-truth profiles are smoothly varying in 2D spaces. It will be good to evaluate proposed methods on more complex scenarios. Also, more details about the compute time for training PINN and PI-RGSM on this dataset can be provided.
3. Limited novelty of work. PI-RGSM seems to be a simple extension of PINN with the inclusion of hard constraints in the outputs. It will be useful to demonstrate if this idea is useful in general on problems outside of groundwater seepage modeling, including benchmark PDEs such as those available in PDEBench and PDEArena.
4. Writing is not clear at several places and could be improved.

**Questions:**

See weaknesses above.

---

### Note · Authors · 2024-11-29

I have read and agree with the venue's withdrawal policy on behalf of myself and my co-authors.